# SYNTHESIZER: RETHINKING SELF-ATTENTION FOR TRANSFORMER MODELS

## ABSTRACT

The dot product self-attention is known to be central and indispensable to state-of-the-art Transformer models. But is it really required? This paper investigates the true importance and contribution of the dot product-based self-attention mechanism on the performance of Transformer models. Via extensive experiments, we find that (1) random alignment matrices surprisingly perform quite competitively and (2) learning attention weights from token-token (query-key) interactions is useful but not that important after all. To this end, we propose SYNTHESIZER, a model that learns synthetic attention weights without token-token interactions. In our experiments, we first show that simple Synthesizers achieve highly competitive performance when compared against vanilla Transformer models across a range of tasks, including machine translation, language modeling, text generation and GLUE/SuperGLUE benchmarks. When composed with dot product attention, we find that Synthesizers consistently outperform Transformers. Moreover, we conduct additional comparisons of Synthesizers against Dynamic Convolutions, showing that simple Random Synthesizer is not only $60\%$ faster but also improves perplexity by a relative $3.5\%$. Finally, we show that simple factorized Synthesizers can outperform Linformers on encoding only tasks.

## 1 INTRODUCTION

Transformer models (Vaswani et al., 2017) have demonstrated success across a wide range of tasks. This has resulted in Transformers largely displacing once popular auto-regressive and recurrent models in recent years. At the heart of Transformer models lies the query-key-value dot product attention. The success of Transformer models is widely attributed to this self-attention mechanism since fully connected token graphs, which are able to model long-range dependencies, provide a robust inductive bias.

But is the dot product self-attention really so important? Do we need it? Is it necessary to learn attention weights via pairwise dot products? This paper seeks to develop a deeper understanding of the role that the dot product self-attention mechanism plays in Transformer models.

The fundamental role of dot product self-attention is to learn self-alignment, i.e., to determine the relative importance of a single token with respect to all other tokens in the sequence. To this end, there have been memory metaphors and analogies constructed to support this claim. Indeed, the terms *query*, *keys*, and *values* imply that self-attention emulates a content-based retrieval process which leverages pairwise interactions at its very core.

Moving against convention, this paper postulates that we cannot only do without dot product self-attention but also content-based *memory-like* self-attention altogether. Traditionally, attention weights are learned at the instance or sample level, where weights are produced by instance-level pairwise interactions. As a result, these instance-specific interactions often fluctuate freely across different instances as they lack a consistent global context.

This paper proposes SYNTHESIZER, a new model that learns to synthesize the self-alignment matrix instead of manually computing pairwise dot products. We propose a diverse suite of synthesizing functions and extensively evaluate them. We characterize the source information that these synthesizing functions receive, i.e., whether they receive information from individual tokens, token-token

interactions, and/or global task information. Intuitively, different source inputs to the synthesizing functions should capture diverse views, which may be useful when employed in conjunction.

Aside from generalizing the standard Transformer model, we show that it is possible to achieve competitive results with fully global attention weights that do not consider token-token interactions or any instance-level (local) information at all. More specifically, a *random* matrix SYNTHESIZER model achieves a 27.27 BLEU score on WMT 2014 English-German[1]. Via a set of rigorous experiments, we observe that the popular and well-established dot-product content-based attention can be approximated with simpler variants such as random matrices or dense layers without sacrificing much performance in some cases.

In our experiments, we also show that our relatively simple Synthesizer models also outperform Dynamic Convolutions (Wu et al., 2019) with a +3.5% relative improvement in perplexity while being 60% faster. On encoding tasks, our factorized Synthesizers can outperform other low-rank efficient Transformer models such as Linformers (Wang et al., 2020).

While simple Synthesizer models are able to perform competitively, our experiments show that the pairwise dot product is still ultimately helpful. When composing our synthesizing functions with dot products, we find that they consistently improve the performance of Transformers. In general, we believe our findings will spur further investigation and discussion about the true role and utility of the self-attention mechanism in Transformer models.

**Our Contributions**   Our key contributions are described as follows:

- We propose Synthetic Attention, a new way of learning to attend without explicitly attending (i.e., without dot product attention or content-based attention). Instead, we generate the alignment matrix independent of token-token dependencies and explore a potpourri of parameterized functions for synthesizing attention matrices.

- We propose SYNTHESIZER, a new model that leverages Synthetic Attention. The model performs competitive to state-of-the-art Transformer models on a wide range of language tasks, including machine translation and language modeling.

- Moreover, we show that (1) random learnable alignment matrices perform competitively and (2) token-token dependencies are not necessary to achieve good performance with Transformer models on certain tasks.

- On large-scale masked language modeling on the C4 dataset (Raffel et al., 2019) and fine-tuning on SuperGLUE and GLUE benchmarks, we show that simple random Synthesizers can outperform/match Lightweight Dynamic convolutions (Wu et al., 2019) along with outperforming Transformers and Universal Transformers (Dehghani et al., 2018). On two encoding tasks, factorized random Synthesizers outperform low-rank Linformers (Wang et al., 2020).

## 2   RELATED WORK

Attention-based models are used across a wide spectrum of problem domains. Such models are especially popular, due to their effectiveness, in the language and vision domains. Attention models can be traced back to the machine translation models of (Bahdanau et al., 2014) and (Luong et al., 2015), where attention is employed to learn soft word alignments between language pairs. The intuition behind the attention mechanism is deeply-rooted in the notion of memory-based retrieval (Graves et al., 2014; Weston et al., 2014), in which soft differentiable addressing of memory was initially proposed.

The paradigm of learning self-alignments, also known as self-attention, has been largely popularized by Transformer models (Vaswani et al., 2017). This technical narrative has also been explored by a number of other recent studies, including those on intra-attention (Parikh et al., 2016), self-matching networks (Wang et al., 2017), and LSTMN (Cheng et al., 2016). To this end, Transformer models, which function primarily based on self-attention and feed-forward layers, generally serve as a reliable replacement for autoregressive recurrent models.

---

[1]The originally reported result is 27.30.

The self-attention layer itself has been the subject of many recent technical innovations. For example, recent studies have investigated improving the layer's overall efficiency via sparsification and reducing the complexity of computing the alignment matrix (Child et al., 2019; Kitaev et al., 2020; Huang et al., 2018; Tay et al., 2020; Beltagy et al., 2020). These methods are tightly coupled with the query-key-value paradigm, employing a form of memory-based content retrieval as an attention mechanism. On the other end of the spectrum, there have been studies that advocate for replacing self-attention with convolution (Wu et al., 2019). The recent surge in interest in simplifying the attention mechanism raises important questions about the role and utility of the pairwise dot products, which are one the defining characteristics of self-attention models. Meanwhile, in the image domain, (Cordonnier et al., 2019) shows connection of Transformers with CNNs.

Our work is a new take on the self-attention mechanism in Transformer models. We delve deeper, starting with replacing the pairwise dot products with what we call synthesizing functions that learn attention matrices that may or may not depend on the input tokens. The most closely related work is ((Raganato et al., 2020)), in which the authors propose using fixed (i.e., not learned) attention patterns in Transformer encoders. However, the scope of their work is limited to encoders and relies on manually defined handcrafted patterns that seem to work well. Our work takes this intuition further and expands on this narrative.

## 3 THE PROPOSED METHOD

This section introduces our proposed SYNTHESIZER model. At its core, our model is essentially a Transformer model with self-attention modules replaced with our Synthetic Attention modules. Figure 3.1 illustrates the key ideas behind (a) Transformer (b) Dense Synthesizers and (c) Random Synthesizers.

### 3.1 SYNTHESIZER MODEL

This section introduces Synthetic Attention, our proposed self-attention module. Our model removes the notion of query-key-values in the self-attention module and directly synthesizes the alignment matrix instead.

**Dense Synthesizer** Let us consider the simplest variation of the SYNTHESIZER model which is conditioned on each input token. Overall, our method accepts an input $X \in \mathbb{R}^{\ell \times d}$ and produces an output of $Y \in \mathbb{R}^{\ell \times d}$. Here, $\ell$ refers to the sequence length and $d$ refers to the dimensionality of the model. We first adopt $F(.)$, a parameterized function, for projecting input $X_i$ from $d$ dimensions to $\ell$ dimensions.

$$B_i = F(X_i) \tag{1}$$

where $F(.)$ is a parameterized function that maps $\mathbb{R}^d$ to $\mathbb{R}^\ell$ and $i$ is the $i$-th token of $X$ and is applied position-wise (to each vector in the sequence of length $\ell$). Intuitively, this can be interpreted as learning a token-wise projection to the sequence length $\ell$. Essentially, with this model, each token predicts weights for each token in the input sequence. In practice, we adopt a simple two layered feed-forward layer with ReLU activations for $F(.)$:

$$F(X_i) = W_2(\sigma_R(W_1(X_i) + b_1)) + b_2 \tag{2}$$

where $\sigma_R$ is the ReLU activation function and $W_1 \in \mathbb{R}^{d \times d}$ and $W_2 \in \mathbb{R}^{d \times \ell}$. Hence, $B_i$ is now of $\mathbb{R}^\ell$. Given $B \in \mathbb{R}^{\ell \times \ell}$, we now compute:

$$Y = \text{Softmax}(B)G(X) \tag{3}$$

where $G(.)$ is another parameterized function of $X$ that is analogous to $V$ (value) in the standard Transformer model. This approach eliminates the dot product attention $Y = \text{Softmax}(QK^\top)V$ altogether by replacing $QK^\top$ in standard Transformers with the synthesizing function $F(.)$.

**Random Synthesizer** The previous variant learns synthetic attention by conditioning on each input of $X$ and projecting to $\ell$ dimensions. Hence, the Dense Synthesizer conditions on each token independently, as opposed to pairwise token interactions in the vanilla Transformer model. We consider another variation of SYNTHESIZER where the attention weights are not conditioned on any

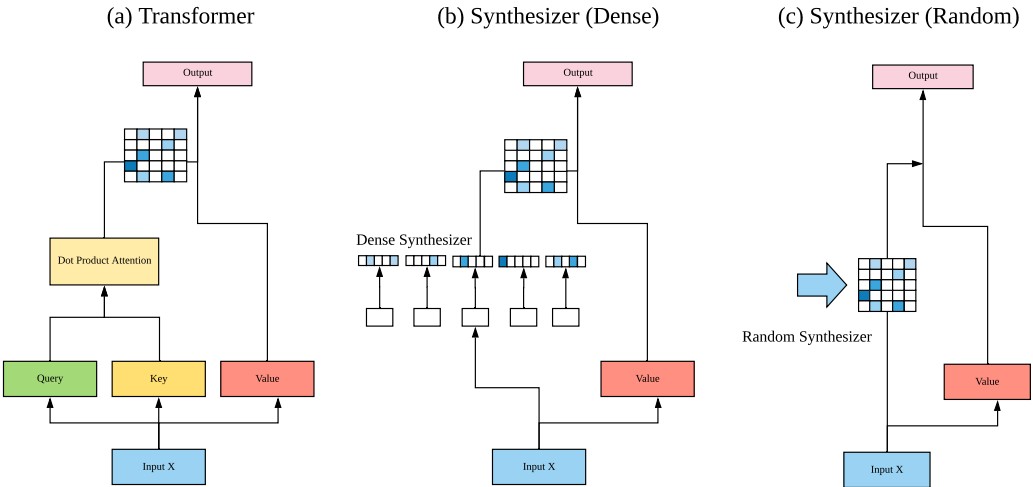

Figure 1: Our proposed SYNTHESIZER model architecture.

input tokens. Instead, the attention weights are initialized to random values. These values can then either be trainable or kept fixed (denoted as *Fixed*).

Let $R$ be a randomly initialized matrix. The Random Synthesizer is defined as:

$$Y = \text{Softmax}(R)G(X). \tag{4}$$

where $R \in \mathbb{R}^{\ell \times \ell}$. Notably, each head adds $\ell^2$ parameters to the network. The basic idea[2] of the Random Synthesizer is to not rely on pairwise token interactions or any information from individual token but rather to learn a task-specific alignment that works well globally across many samples. This is a direct generalization of the recently proposed fixed self-attention patterns Raganato et al. (2020).

**Factorized Models**    The Dense Synthesizer adds $d \times \ell$ parameters to the network. On the other hand, the Random Synthesizer adds $\ell \times \ell$ parameters. Here, note that we omit the $Q, K$ projections in the standard Transformer which results in further parameter savings. Despite these savings, synthesized models can be cumbersome to learn when $\ell$ is large. Hence, we propose factorized variations of the SYNTHESIZER models and show that these variants perform comparably in practice.

**Factorized Dense Synthesizer**    Factorized outputs not only slightly reduce the parameter cost of the SYNTHESIZER but also aid in preventing overfitting. The factorized variant of the dense synthesizer can be expressed as follows:

$$A, B = F_A(X_i), F_B(X_i) \tag{5}$$

where $F_A(.)$ projects input $X_i$ into $a$ dimensions, $F_B(.)$ projects $X_i$ to $b$ dimensions, and $a \times b = \ell$. The output of the factorized module is now written as:

$$Y = \text{Softmax}(C)G(X). \tag{6}$$

where $C = H_A(A) * H_B(B)$ where $H_A, H_B$ are tiling functions and $C \in \mathbb{R}^{\ell \times \ell}$. The tiling function simply duplicates the vector $k$ times, i.e., $\mathbb{R}^\ell \to \mathbb{R}^{\ell \times k}$. In this case, $H_A(\cdot)$ is a projection of $\mathbb{R}^a \to \mathbb{R}^{a \times b}$ and $H_B(\cdot)$ is a projection of $\mathbb{R}^b \to \mathbb{R}^{b \times a}$. To avoid having similar values within the same block, we compose the outputs of $H_A$ and $H_B$.

**Factorized Random Synthesizer**    Similar to Factorized Synthesizers, we are also able to factorize $R$ into low rank matrices $R_1, R_2 \in \mathbb{R}^{\ell \times k}$.

$$Y = \text{Softmax}(R_1 R_2^\top)G(X). \tag{7}$$

---

[2]We were not expecting this variation to work at all, but it turns out to be a strong baseline.

Therefore, it is easy to see that, for each head, this reduces the parameter costs from $\ell^2$ to $2(\ell k)$ where $k << \ell$ and hence helps prevent overfitting. In practice, we use a small value of $k = 8$.

**Mixture of Synthesizers**   Finally, we note that all of the proposed synthetic attention variants can be mixed in an additive fashion. This can be expressed as:

$$Y = \text{Softmax}(\alpha_1 S_1(X) + \cdots \alpha_N S_N(X))G(X). \tag{8}$$

where $S(.)$ is a parameterized synthesizing function and the $\alpha$ (where $\sum \alpha = 1$) are learnable weights. In the case of mixing Random Factorized with standard Dense Synthesizers, this is expressed as:

$$Y = \text{Softmax}(\alpha_1 R_1 R_2^\top + \alpha_2 F(X))G(X). \tag{9}$$

We investigate several Mixture of Synthesizers variants in our experiments.

**On Parameters Depending on Sequence Length**   Random and dense Synthesizers both rely on parameters that depend on length $\ell$. In general, we define a maximum length and dynamically truncate to the actual length of each batch. We note that this is in similar spirit to trainable positional encodings which have been common practice in Transformer models. Hence, we do not forsee any issue here. In the case that this is really a problem, one potential solution is to project to a smaller value $b$ and tile $b$ to the maximum sequence length. We leave this exploration to future work.

## 3.2   DISCUSSION

This paper asks fundamental questions about the attention matrix $A$ and whether it is possible to synthesize $A$ by alternate means other than pairwise attention. It is worth noting that the regular dot product attention can also be subsumed by our SYNTHESIZER framework, i.e., SYNTHESIZER generalizes the Transformer model. In the case of the Transformer, the synthesizing function in question is $S(X) = F_Q(X)F_K(X)^\top$.

| Model | $S(X)$ | Condition On | Sample | Interact | $|\theta|$ |
|---|---|---|---|---|---|
| Dot Product | $F_Q(X)F_K(X_i)^\top$ | $X_j\ \forall j$ | Local | Yes | $2d^2$ |
| Random | $R$ | N/A | Global | No | $.\,\ell^2$ |
| Fac. Random | $R_1 R_2^\top$ | N/A | Global | No | $2\ell k$ |
| Dense | $F_1\sigma(F_2(X_i))$ | $X_i$ | Local | No | $d^2 + d\ell$ |
| Fac. Dense | $H_A(F_A(X_i))) * H_B(F_B(X_i)))$ | $X_i$ | Local | No | $d^2 + d(k_1 + k_2)$ |

Table 1: Overview of all Synthesizing Functions.

Table 1 lists the different model variants explored within our SYNTHESIZER framework. The 'condition on' column refers to whether the synthesized output is produced as a function of $X_i$ or every $X_i, X_j$ pair. The 'sample' column indicates whether a given variant leverages local or global context. Random Synthesizers are global because they share the same global alignment patterns across all samples. Dense Synthesizers are considered to be local as they are conditioned on $X_i$, which makes the alignment pattern dependent on each individual sample. To this end, it is imperative for synthesized models to have multiple heads to be effective.

## 4   EXPERIMENTS

This section outlines our experimental setup and results. We first conduct experiments on five tasks to evaluate the effectiveness[3] of different Synthesizer variants along with how they compare to the vanilla Transformer. Specifically, we conduct experiments on (1) machine translation (EnDe, EnFr) (2) autoregressive language modeling (LM1B) (3) text generation (summarization and dialogue modeling and (4) multi-task natural language processing (GLUE/SuperGLUE). Details of each experiments can be found in the appendix.

---

[3]Note that we are primarily interested in making controlled comparisons instead of going for the state-of-the-art result on each task.

**Notation of Variants** We use R to denote Random, D to denote Dense and V to denote vanilla dot product attention. Fix to represent Fixed Random, FR to represent Factorized Random and FD to represent Factorized random. For Mixture Synthesizers, we use + to denote that two methods are mixed.

## 4.1 COMPARING SYNTHESIZER VARIANTS AND TRANSFORMER MODELS

This section dives into a detailed study of multiple Synthesizer variants and the base Transformer model.

| Model | NMT (BLEU) | | | LM (PPL) | |
| --- | --- | --- | --- | --- | --- |
| | $|\theta|$ | EnDe | EnFr | $|\theta|$ | LM |
| Trans.[†] | 67M | 27.30 | 38.10 | - | - |
| Trans. | 67M | 27.67 | 41.57 | 70M | 38.21 |
| **Synthesizer Models** | | | | | |
| Fix | 61M | 23.89 | 38.31 | 53M | 50.52 |
| R | 67M | 27.27 | 41.12 | 58M | 40.60 |
| FR | 61M | 27.30 | 41.12 | 53M | 42.40 |
| D | 62M | 27.43 | 41.39 | 53M | 40.88 |
| FD | 61M | 27.32 | 41.57 | 53M | 41.20 |
| R+D | 67M | 27.68 | 41.21 | 58M | 42.35 |
| D+V | 74M | 27.57 | 41.38 | 70M | **37.27** |
| R+V | 73M | **28.47** | **41.85** | 70M | 40.05 |

Table 2: Experimental Results on WMT'14 English-German, WMT'14 English-French Machine Translation tasks and Language Modeling One Billion (LM1B). † denotes original reported results in (Vaswani et al., 2017).

**Experimental Results on MT/LM** First, we observe that our Random Synthesizer baseline achieves 27.27 on EnDe and 41.12 on EnFr. The non-trainable (i.e., fixed) variant performs substantially worse, but still yields surprisingly strong ≈ 24 BLEU with fixed random attention weights. Most other SYNTHESIZER variants achieve competitive performance, although with slight performance degradation compared to Transformers. An interesting finding is that the Mixture model of Random + Dense synthesizer performs comparably to vanilla Transformers on EnDe. When mixing the standard dot product attention, performance further increases by +0.8 BLEU points (EnDe). In general, the performance of SYNTHESIZER variants are competitive with Transformers for this task. On LM1b, We find that the Random Synthesizers perform within 1-2 PPL points away from the vanilla Transformer model. The best performing model is the Synthesizer (D+V), which achieves the best performance on this setting.

**Results on Text Generation** For summarization, we find that the (R) and (D) variants do not outperform Transformers. The performance of the (D) model is ≈ 2 Rouge-L points below Transformers. Hence, we postulate that the local sample-wise pairwise interactions are important for the summarization task. On the other hand, the utility of synthesized attention can also be observed, i.e., the (R+V) and (R+D) models both outperform Transformers. On the dialogue task, Synthesizers (R) and (D) both outperform vanilla Transformers by a reasonable margin (≈ 1-3) points across most/all metrics. The best performing model here is the (D) variant. Surprisingly, unlike most other tasks, the (+V) variants do not perform well, signifying that dot product self-attention may actually be harmful for this task.

| Model | Sum. | Dialogue | | |
| --- | --- | --- | --- | --- |
| | RL | $B_4$ | RL | Met. | CIDr |
| Trans. | 35.77 | 3.20 | 13.38 | 5.89 | 18.94 |
| **Synthesizer Models** | | | | | |
| R | 33.10 | 2.25 | 15.00 | 6.42 | 19.57 |
| D | 33.70 | **4.02** | **15.22** | **6.61** | **20.54** |
| D+V | **36.02** | 3.57 | 14.22 | 6.32 | 18.87 |
| R+V | 35.95 | 2.28 | 14.79 | 6.39 | 19.09 |

Table 3: Experimental results on Abstractive Summarization (CNN/Dailymail) and Dialogue Generation (PersonaChat). We report on RL (Rouge-L), B4 (Bleu-4), Met. (Meteor) and CIDr.

## 4.2 COMPARING SYNTHESIZERS WITH DYNAMIC CONVOLUTIONS

To ascertain the competitiveness of Synthesizers, we also compare them with Dynamic convolutions (Wu et al., 2019). We compare them on (1) pretraining perplexity using the masked language modeling objective on C4 and (2) downtream finetuning results on GLUE and SuperGLUE.

**Results on Masked Language Modeling** We also benchmark the speed of these models. In order to do so, we conduct additional experiments on the T5 adaptation of masked language modeling on the C4 dataset (Raffel et al., 2019) by comparing against lightweight dynamic convolutions (Wu et al., 2019) on a masked language modeling task. We also take this chance to benchmark the

| Model | Glue | CoLA | SST | MRPC | STSB | QQP | MNLI | QNLI | RTE |
|---|---|---|---|---|---|---|---|---|---|
| T5 (Base) | 83.5 | 53.1 | **92.2** | **92.0/88.7** | 89.1/88.9 | 88.2/91.2 | 84.7/**85.0** | 91.7 | 76.9 |
| T5 (Base+) | 82.8 | 54.3 | 92.9 | 88.0/83.8 | 85.2/85.4 | 88.3/91.2 | 84.2/84.3 | 91.4 | 79.1 |
| DyConv | 69.4 | 33.9 | 90.6 | 82.6/72.5 | 60.7/63.1 | 84.2/88.2 | 73.8/75.1 | 84.4 | 58.1 |
| Syn (R) | 75.1 | 41.2 | 91.2 | 85.9/79.4 | 74.0/74.3 | 85.5/89.0 | 77.6/78.1 | 87.6 | 59.2 |
| Syn (D) | 72.0 | 18.9 | 89.9 | 86.4/79.4 | 75.3/75.5 | 85.2/88.3 | 77.4/78.1 | 86.9 | 57.4 |
| Syn (D+V) | 82.6 | 48.6 | 92.4 | 91.2/87.7 | 88.9/89.0 | 88.6/91.5 | 84.3/84.8 | 91.7 | 75.1 |
| Syn (R+V) | **84.1** | **53.3** | **92.2** | 91.2/87.7 | **89.3/88.9** | **88.6/91.4** | **85.0**/84.6 | **92.3** | **81.2** |

Table 5: Experimental results (dev scores) on multi-task language understanding (GLUE benchmark) for *small* model and `en-mix` mixture. Note: This task has been co-trained with SuperGLUE.

| Model | SGlue | BoolQ | CB | CoPA | MultiRC | ReCoRD | RTE | WiC | WSC |
|---|---|---|---|---|---|---|---|---|---|
| T5 (Base) | 70.3 | 78.2 | 72.1/83.9 | 59.0 | 73.1/32.1 | **71.1/70.3** | 77.3 | **65.8** | **80.8** |
| T5 (Base+) | 70.7 | 79.3 | 81.1/87.5 | 60.0 | 75.1/34.4 | 71.7/70.7 | 80.5 | 64.6 | 71.2 |
| DyConv | 57.8 | 66.7 | 65.9/73.2 | 58.0 | 57.9/8.71 | 58.4/57.4 | 69.0 | 58.6 | 73.1 |
| Syn (R) | 61.1 | 69.5 | 54.6/73.2 | 60.0 | 63.0/15.7 | 58.4/57.4 | 67.5 | 64.4 | 66.3 |
| Syn (D) | 58.5 | 69.5 | 51.7/71.4 | 51.0 | 66.0/15.8 | 54.1/53.0 | 67.5 | 65.2 | 58.7 |
| Syn (D+V) | 69.7 | 79.3 | 74.3/85.7 | 64.0 | 73.8/33.7 | 69.9/69.2 | 78.7 | 64.3 | 68.3 |
| Syn (R+V) | **72.2** | **79.3** | **82.7/91.1** | **64.0** | **74.3/34.9** | 70.8/69.9 | **82.7** | 64.6 | 75.0 |

Table 6: Experimental results (dev scores) on multi-task language understanding (SuperGLUE benchmark) for *small* model and `en-mix` mixture. Note: This task has been co-trained with GLUE.

speed of Synthesizers compared with Transformers. Experiments are conducted on Mesh Tensorflow (Shazeer et al., 2018) and ran on 2x2 TPU V3 Chips for approximately $524K$ steps.

| Model | Log PPL | Steps/Sec | Params | FLOPS |
|---|---|---|---|---|
| Transformer (Vaswani et al., 2017) | 1.865 | 3.90 | 223M | $3.70 \times 10^{12}$ |
| Dynamic Conv (Wu et al., 2019) | 2.040 | 2.65 | 257M | $3.93 \times 10^{12}$ |
| Lightweight Conv (Wu et al., 2019) | 1.972 | 4.05 | 224M | $3.50 \times 10^{12}$ |
| Synthesizer (D) | 1.965 | 3.61 | 224M | $3.80 \times 10^{12}$ |
| Synthesizer (R) | 1.972 | **4.26** | 254M | $3.36 \times 10^{12}$ |
| Synthesizer (R+V) | 1.849 | 3.79 | 292M | $4.03 \times 10^{12}$ |
| Synthesizer (D+V) | **1.832** | 3.34 | 243M | $4.20 \times 10^{12}$ |

Table 4: Validation perplexity scores on C4 dataset (Raffel et al., 2019). All models are at approximately similar parameterization.

**Results on MLM** Table 4 reports the validation set log perplexity on masked language modeling[4]. We observe that Synthesizers (R) can outperform Dynamic Convolutions by a relative +3.5% while being +60% faster. Against Lightweight Dynamic Convolutions, we match the performance while being +5% faster. Given that this is the simple random Synthesizer baseline, we find this extremely interesting how it is able to outperform dynamic convolutions, a relatively complex model. The Random Synthesizer also has less FLOPS compared to both convolution models. On the other hand, the Mixture Synthesizer models that use the dot product attention improves the performance of the base Transformer model with relatively an equal model speed. Finally, similar to the earlier results, we see a consistent performance gain of Synthesizer (D+V) and Synthesizer (R+V) outperforming the base Transformer model.

**Results on GLUE and SuperGLUE** Tables 5 and 6 report results on the GLUE and SuperGLUE benchmarks. We note that the (R) and (D) variants of SYNTHESIZER do not achieve reasonable performance. This can be largely attributed to the fact that the encoder self-attention in the T5 setting also functions as a cross-sentence attention. For example, in the entailment or reading comprehension tasks, the premise and hypothesis are concatenated together and self-attention effectively acts as cross-sentence attention[5]. On datasets like SST, a straightforward sentiment classification

---

[4]Note that this follows the sequence transduction style in T5.

[5]On a related note, the perceived success of pairwise self-attention might also be attributed to the fact that these public benchmarks are bias towards pairwise matching tasks. In reality, this is computationally prohibitive for many practical real-world applications (Seo et al., 2018).

task, this cross sentence attention is not necessary and therefore Syn (R) and Syn (D) both perform competitively. To this end, Dynamic Convolutions (Wu et al., 2019) also do not have this encoder "cross-attention" and therefore also suffer on many of these pairwise matching tasks. Notably, in this 'no cross attention' setting, the Random Synthesizers are are 4 to 5 percentage points higher in GLUE/SuperGLUE score compared to Dynamic Convolutions.

Optimistically, we observe that the mixture model Syn (R+V) outperforms the T5 model by a substantial margin (+1.9 points on SuperGLUE and +0.6 points on GLUE). Naturally, the hybrid mixture model also very substantially outperforms Dynamic Convolution. Finally to ensure that the Syn (+V) variations are not outperforming Transformers due to simply having more parameters, we also compared with T5 (Base+) which has equal number of parameters to Syn (+V) variants (approximately $\approx 10M$ more parameters). Our results show that Synthesizers (+V) still outperform T5 (Base+).

### 4.3 COMPARING SYNTHESIZERS WITH LINFORMERS

We conduct more experiments comparing factorized random Synthesizers with Linformers. Since Linformer cannot be used to decode, we compare them on two encoding tasks from tensorflow datasets (AGnews (Zhang et al., 2015) and movie reviews (Maas et al., 2011)). We use $k$=32 for both factorized models. We also benchmark Transformers on this task. Note we do not use contextualized embeddings so results are not comparable with other work.

| Model | News | Reviews | Steps/Sec |
|---|---|---|---|
| Transformer | 88.83 | 81.34 | 1.09 |
| Linformer | 86.50 | 82.86 | 1.09 |
| Syn (FR) | 86.53 | 83.39 | **1.10** |
| Syn (FR+V) | **89.13** | **84.61** | 0.80 |

Table 7: Results on Encoding only tasks (accuracy).

**Results** We notice that factorized Synthesizers (FR) are competitive with Linformers and Transformers on this task. The accuracy of Syn (FR) is competitive with Linformers while Syn (FR+V) outperforms both Transformers and Linformers.

### 4.4 OVERALL SUMMARY OF QUANTITATIVE RESULTS

This section summarizes our overall findings.

- **Synthetic Attention is competitive even without Dot Product Attention** On all evaluated tasks, we showed that synthesized attention functions competitively, i.e., it achieves performance reasonably close to the dot product self-attention. On one task (dialogue generation), the dot product self-attention is found to actually degrade performance. Amongst the other tasks, machine translation is the least affected by the removal of the vanilla dot product. These findings allow us to introspect about whether pairwise comparisons for self-attention are even necessary. On the multi-task language understanding benchmark, the self-attention functions as a form of cross-attention by concatenating sentence pairs. Hence, synthesize attention performance is considerably worse than vanilla Transformers.

- **Synthetic Attention and Dot Product Attention are highly complementary** Overall, we also observe that the dot product attention is very helpful. To this end, synthetic attention is highly complementary to the pairwise dot product attention. While Synthetic Attention can usually achieve competitive and fast performance on its own, synthetic attention boosts performs, composing multiple synthetic attention (and dot product attention) together shows gains on almost all tasks that we have investigated. Hence, we believe this to be a robust finding.

  **The simplest Synthesizers such as Random Synthesizers are fast competitive baselines** Finally, we note that simple random Synthesizers are competitive with dynamic convolutions and Linformers, which are recently proposed models. On two encoding task and a large-scale masked language modeling task, we show that random (or factorized random) Synthesizers remain competitive to other fast or efficient Transformer models.

# 5 CONCLUSION

This paper proposed SYNTHESIZER, a new Transformer model that employs Synthetic Attention. We conducted a principled study to better understand and evaluate the utility of global alignment and local, instance-wise alignment (e.g., independent token and token-token based) in self-attention. We show that, on multiple tasks such as machine translation, language modeling, dialogue generation, masked language modeling and document classification, synthetic attention demonstrates competitive performance compared to vanilla self-attention. Moreover, for the dialogue generation task, pairwise interactions actually hurt performance. Notably, we reemphasize that this study refers to self-attention. We found that we are not able to replace cross-attention with simpler variants in most cases. Via a set of additional large-scale experiments, also find that Synthesizers can outperform or match Dynamic Convolutions and Factorized Synthesizers can outperform other low rank Linformer models.

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

## A  APPENDIX

### A.1  DETAILED SETUP FOR EXPERIMENTS

**Machine Translation**   We implement our models in Tensor2Tensor, using the standard **base** hyper-parameter settings. Specifically, we use byte-pair encoding (BPE), 6-layered Transformer networks with hidden size 512, filter size of 2048 and 8 heads. We use label smoothing of 0.1. The maximum sequence length is set to 256. Training is performed using 8 x V100 GPUs. We train all models for $250K$ steps and report results at the last checkpoint. We use a length penalty of 0.6 and beam size of 4 following the default settings. We also compare with standard Transformer models. In the interest of keeping a consistent, fair evaluation across all model settings, we do not use checkpoint averaging or tune the decoding hyperparameters although this generally leads to better performance. We evaluate BLEU scores using `sacrebleu`.

**Language Modeling**   We implement our models in Tensor2Tensor using the packed TPU setup of sequence length 256. We train our models on $300K$ steps on 16 TPU V2 chips. We use the `lmx_base` model setting for fair comparison across all model variations. The model has 6 layers and 8 heads, along with a filter width of 2048 and hidden size of 512. We used `conv_relu` for the positional feed-forward layers across all baselines since we find them to perform slightly better. We report results (subword level perplexity scores) on the test set at the final checkpoint.

**Summarization**   For the summarization task, we train all models for $300K$ steps and a batch size of 128. All models use the *base* size setting. For the dialogue generation task, due to the smaller dataset size, we train a *small* model for $20K$ steps. All results are reported on the test set. For the summarization task, we use the well-established metrics, i.e., Rouge-1, Rouge-2 and Rouge-L. Experiments are conducted using Mesh Tensorflow.

**Dialogue Generation**   For the dialogue generation task, we train our models on the small size for $20K$ steps. Experiments are conducted in Tensor2Tensor. We use NLG-Eval[6] (Sharma et al., 2017) and report BLEU-1, BLEU-4, Rouge-L, Meteor, CIDr and Embedding based similarity scores (Emb).

**Multi-Task Language Understanding**   Our experiments are based on the T5 repository[7] implemented in Mesh Tensorflow (Shazeer et al., 2018). We pre-train the vanilla T5 models and our models for 524288 steps using the span denoising objective. We then co-train the model on multiple tasks. We co-train on the en_mix mixture (SuperGLUE and GLUE) for $100k$ steps with a constant learning rate of $10^{-3}$. Embedding and Softmax output layer parameters are kept fixed. The maximum sequence length is set to 512. We evaluate on the en_mix mixture as defined in the original codebase which is comprised of training GLUE, SuperGLUE and SQuAD in a single model.

**Pretraining experiments on C4**   Experiments are conducted on Mesh Tensorflow. We pretrain for 524288 steps and report the perplexity on the validation set. We use 2x2 TPU V3 chips for our experiments. The sequence length is 512 and optimizer is Adafactor.

**Experiments on Document Classification**   We run experiments in JAX/FLAX (`https://github.com/google/flax`) with base size models of 8 heads, 6 layers, MLP dimensions of 2048 and a hidden size of 512. We use the Adam optimizer with learning rate 0.05 and $8K$ steps linear warmup. We train for $10K$ steps and report evaluation results at $10K$ step. We use a batch size of 128. We build a new sentencepiece model for each new dataset comprising of $32K$ tokens. No pretraining or contextualized embeddings are used. Experiments are run on 16 TPU v3 chips.

## A.2   ADDITIONAL VARIANTS OF SYNTHESIZER

We report results of several additional variants of SYNTHESIZER, most of which we found to have marginal or no improvement over the simple dense/random variations.

- Convolution - Applying a 1D convolution instead of a 2 layer nonlinear network. We vary the filter width in our experiments.

- Bottleneck - Converting the 2 layered feed forward network to a bottleneck layer, e.g., $512 \rightarrow 16 \rightarrow 512$. We also experiment with a convolutional variant of bottleneck, i.e., projecting to low dimension space and then projecting back to high dimensions.

- Gated Linear Units (GLU), applying the GLU units of (Dauphin et al., 2017) as the Synthesizing function.

---

[6] `https://github.com/Maluuba/nlg-eval`.
[7] `https://github.com/google-research/text-to-text-transfer-transformer`

| Variant | BLEU |
|---|---|
| Transformer | 27.67 |
| Random | 27.27 |
| Dense | 27.43 |
| Conv ($f = 3$) Linear | 27.43 |
| ConvReluConv ($f = 3$) | 27.51 |
| ConvReluConv ($f = 5$) | 27.56 |
| ConvReluConv ($f = 3, 5$) | 27.49 |
| Bottleneck + Dense | 27.43 |
| Bottleneck + ConvReluConv | 27.72 |
| GLU | 27.43 |

Table 8: Results for additional SYNTHESIZER variants on WMT EnDe (BLEU scores)

## A.3 EFFECT OF NUMBER OF HEADS

We also investigate the impact of the number of heads on performance. We trained three Random Synthesizer models for the `small` version of the machine translation tasks using the T5 framework without pretraining. For simplicity, evaluation is done via greedy decoding. We report scores on the development set. We are mainly interested in relative performance and not absolute numbers. Table 9 reports the results on varying the number of heads on performance.

| Heads | EnDe | EnFr | EnRo |
|---|---|---|---|
| Syn $h$=2 | 19.43 | 34.12 | 18.67 |
| Syn $h$=4 | 20.42 | 35.26 | 19.78 |
| Syn $h$=8 | 20.88 | 34.92 | 20.28 |
| Syn $h$=16 | 21.71 | 35.26 | 20.43 |
| Syn $h$=32 | **21.72** | **36.01** | **20.52** |

Table 9: Effect of number of heads on multi-task MT. Increasing the number of heads improves performance.

## A.4 ANALYSIS

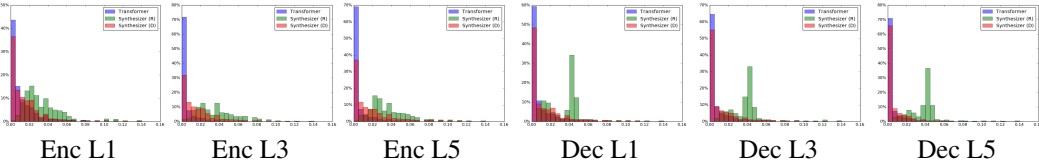

| Enc L1 | Enc L3 | Enc L5 | Dec L1 | Dec L3 | Dec L5 |

Figure 2: Histogram of Encoder and Decoder Attention Weights on MT (WMT EnDe). L denotes the layer number and Enc/Dec denotes encoder or decoder.

**Distribution of Weights** We are interested in investigating how the synthetically generated attention weights differ from the dot product attention weights. Figure 2 shows the attention histograms on trained Transformer and SYNTHESIZER models. We report histograms at layers 1, 3, and 5 of a 6 layered (Transformer or SYNTHESIZER) model at $50K$ steps. We found that the weight distributions remain relatively identical thereafter. Figure 3 shows the initialization state. We observe that there are distinct differences in the weight distribution of SYNTHESIZER and Transformer models. The variance of the SYNTHESIZER weights tends to be higher. On the other hand, the weights on the Transformer model tends to gravitate near 0 and have smaller variance. There are also notable differences across the (R) and (D) SYNTHESIZER variants. Specifically, the (D) model in general has greater max values with more values in the 0.1-0.2 range while the values of the $R$ model tends to stay closer to 0.

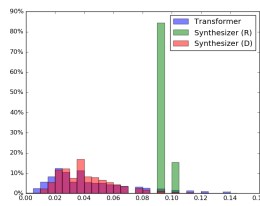

Figure 3: Init Decoder weights (Reference)

## A.5    WHAT PATTERNS DO SYNTHESIZERS LEARN?

In this section, we perform a deeper analysis of the SYNTHESIZER model.

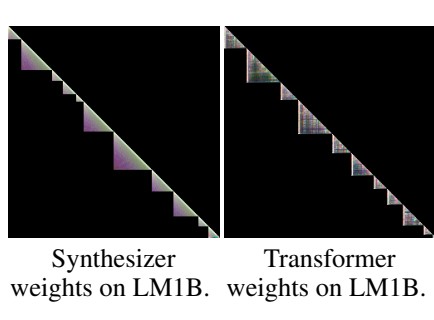

Synthesizer      Transformer
weights on LM1B.  weights on LM1B.

**Analysis** Finally, we are interested to understand what these Synthesizer models are learning. We inspect the random synthetic attention weights for language modeling task LM1B and visualise the differences compared to the vanilla attention. We find that, for the LM task, Synthesizers are capable of learning a local window, emulating the vanilla Transformer quite closely despite starting from completely random. The weights, however, seem smoother and less coarse as compared to the Transformer. This seems to reflect what we expect since the Synthesizer does not benefit from token specific information. We provide additional analysis and visualisation of weights for the Machine Translation task in the supplementary material.

## A.6    MORE ATTENTION WEIGHTS ANALYSIS

This section illustrates the attention weights extracted from different variants of Synthesizer on the machine translation (En-De) task. Weights are extracted from lower layers although we do not find any substantial difference in the patterns in early layers and deeper layers. We extract them from Tensorboard midway during training.

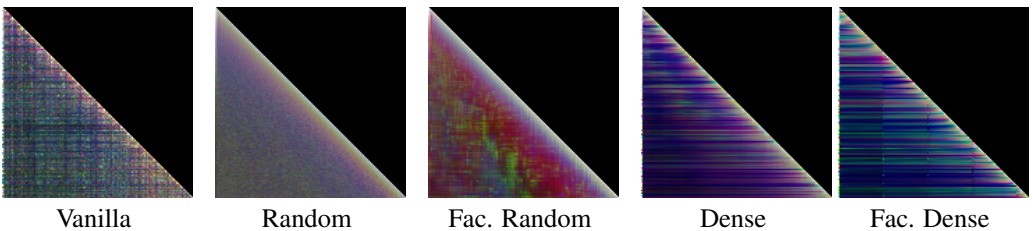

Vanilla      Random      Fac. Random      Dense      Fac. Dense

Figure 4: Visual analysis of Synthetic Attention (decoder) on WMT EnDe.

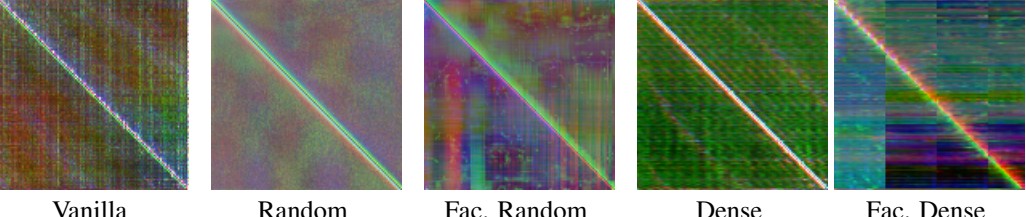

| Vanilla | Random | Fac. Random | Dense | Fac. Dense |

Figure 5: Visual analysis of Synthetic Attention (encoder) on WMT EnDe.

**Analysis** We first observe that these weights differ a lot from the LM weights shown in the main paper in Section 4.5. This shows that the Synthesizer learns very different weights for different tasks. Next, based on the weighs on MT, we observe a very different pattern in all variants of Synthesizer. For the decoder weights, the main difference seems to be the overall magnitude and distribution values of the weights. However, we can easily observe the cracks and lines of the factorized variants. For the encoder weights, we observe that the Random and Dense variants are more uniform. On the other hand, there appears to be structural/regional clustering of values in the factorized variants.

## A.7 CONVERGENCE OF SYNTHESIZERS VS TRANSFORMERS

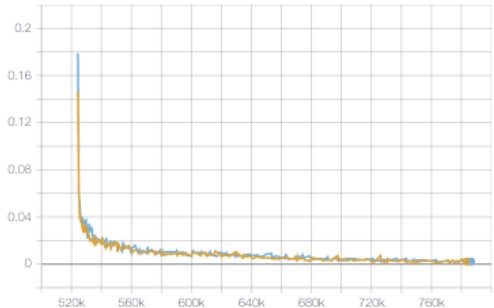

Convergence of Synthesizers (Brown) vs Transformers (Blue)

Figure A.7 shows the convergence of Synthesizers vs Transformers on SuperGLUE finetuning. In general, Synthesizers and Transformers have pretty similar convergence patterns and this is also observed across other tasks.

