# OpenReview forum: "Synthesizer: Rethinking Self-Attention for Transformer Models"
_ICLR.cc/2021/Conference — Reject_

### Official Review · AnonReviewer2 · 2020-10-17
**A rather surprising result which questions the necessity of the standard dot-product attention in Transformers**

**Rating:** 7
**Confidence:** 4

**Review:**

Summary
---------------------
This paper questions the necessity of the standard query-key attention in Transformer layers. It shows that replacing the standard attention mechanism with (1) random (which are learned in a task-specific way), (2) dense (which only depend on the contextualized representation $X_i$ instead of $X_i, X_j$ as usual) attention weights performs almost as well as standard methods. Combining this approach with standard attention via a mixture approach even improves upon the usual dot product attention. I found the results rather surprising, and believe that this would be an interesting and worthwhile contribution to the conference.

Strengths
---------------------
- Interesting set of baselines/experiments (random/dense, factored, mixtures, etc.) with appropriate ablations across the various modeling choices.
- Comprehensive evaluation across various settings: machine translation, language modeling, summarization, dialogue modeling,  and GLUE.
- Very nice analysis of the learned model/attention in the Appendix. I found some of these fascinating so I encourage the authors to consider including them in the main paper.
- The paper is very well written and was a pleasure to read.

Weaknesses
---------------------
- While interesting, I am not sure what the practical computational benefit of this approach seems to be, since the query/key projection component of the Transformer does not take up much time.
- While the point of the paper is not to achieve SoTA performance, there is still nontrivial degradation in performance for many tasks if just using the random/dense approach.

Questions
---------------------
- How much of the performance loss is due to longer sequences being less frequently encountered in the training set? (I.e. the $l$-th dimension of $F(X)$ gets worse for increasing $l$). Have you tried an alternative where you fix use the same $l$ for $l$ greater than (for example) 40?
- Another way to achieve attention that only conditions on $X_i$ (as in the dense approach) would be to train with key/query as usual, and then use the average logits for token $X_i$ (averaged across the training set, for example). Have you considered this approach as a baseline?
- The computational benefit of this approach is somewhat lessened by the fact that one still needs to perform the dense projection for each $X_i$ (for the dense approach at least). This could be avoided if $F(X_i)$ only depends on the non-contextualized representation of the $i$-th word, which means one could precompute $F(X_i)$ for all words in the vocab after training for faster inference. Given that the random baseline works, I imagine this baseline would work as well. Have you considered this approach?

-----------------
EDIT after rebuttal: Thank you for the response.

---

> ### Author Response · Authors · 2020-11-21
> **Response to Review**
>
> Thanks for the great review and extremely insightful comments.
>
> We are happy you enjoyed and appreciated our paper.
>
> Regarding the performance issue on many tasks, the reason for random/dense Synthesizers not doing well on some tasks in GLUE/SuperGLUE is because of the lack of cross attention inductive bias in the encoder and not self-attention per se. We mentioned this in the paper. We felt this was worth including this result in the paper as it provides a balanced view (strengths and weaknesses) of each variant accurately. That said, results on Machine translation, LM, Masked LM, Text generation all show that Random/Dense approaches can do competitively. If anything, the reason for the lack of performance on Glue/SuperGLUE is because of cross attention. In our rebuttal revision, we evaluated another strong baseline (Dynamic Convolutions, which are also another popular  self-attention alternative baseline) and find that the random/dense approach outperforms it on Glue/SuperGLUE. Dynamic Convs also do not have the cross attention inductive bias in the encoder.
>
> Regarding the computational benefit, the random synthesizers presented are the fastest of the proposed variants (this is about 10% faster, which accounts for not needing to compute QKs). We agree that there are many other (orthogonal) approaches targeted at other parts of the Transformers (i.e., MLPs) that can make Transformers faster. To this end, we expect this 10% gain to stack with other approaches so its a nice win for tasks when random synthesizers can do sufficiently well.
>
> Thanks for the questions! They were very interesting and insightful.
>
> Regarding length, this is an interesting question and it is absolutely a great idea that we could “tile the blocks” and use the same block for greater L values. We will try this approach! Thanks for the suggestion.
>
> Regarding the average logits attention, the idea of using the average logits for X_i is indeed very interesting. We think this would make interesting follow up work.
>
> Regarding the precomputed attention, this is indeed a very interesting idea. We have not tried this but we are sincerely interested in exploring this as this would enable a form of “universal attention weights” that can even be used and shared across many tasks. Thanks for the great suggestion!
>
> Thanks for taking the time to review our paper and the many wonderful suggestions.

---

### Official Review · AnonReviewer1 · 2020-10-28
**Needs work**

**Rating:** 4
**Confidence:** 4

**Review:**

Summary:
This paper proposes replacing/combining Transformer self-attention with synthetic attention weights that do not rely on pairwise dependencies between token positions. Synthetic attention relies on either the input at the given position (dense synthesizer) or is altogether randomly initialized (random synthesizer).

The goal of the paper is to show that synthetic attention is a competitive alternative to self-attention.

Some thoughts:
1. The claim that combining synthetic attention with self-attention improves performance seems pretty unfair since the synthetic attention adds extra parameters that the baseline using only self-attention doesn’t have. This claim seems pervasive throughout the experiments but doesn’t appear to be well supported. Also, it would be interesting to see what the learned weights for the different types of attention look like - does it mostly just use self-attention?
2. The fact that synthetic attention is competitive is interesting, but how does it compare to other methods for replacing self-attention (eg: Wu et al, 2019’s work on convolutions)? This question hasn’t really been properly addressed. They report a wide range of experiments in their paper, but the only setting compared here is T5 pre-training, and even there the metric is dev set loss for masked language modeling, which doesn’t objectively mean much.
3. The random synthesizer results are interesting and surprising. It would be interesting to delve deeper into figuring out why the randomly initialized parameters serve as a reasonable proxy for token dependent weights. Are there any results from the Fixed random synthesizer in the paper? Don’t see any.
4. Speed is recorded in terms of FLOPS for T5 pre-training, what about model convergence? Does it take longer for the model to converge? Doesn’t have to be on the T5 pre-training, MT or LM or the other tasks are fine too.
5. Overall, the paper isn’t very clearly written. The motivations are not highlighted - why should one want to use synthetic attention? The notation, language and naming conventions are also a bit sloppy and inconsistent.

It might be worth considering an analytic framing for the paper where the goal is to study what makes the pairwise interactions replaceable and what going from all pairs to convolutions to dense and then finally random looks like in terms of model behavior and outputs. Also, is self-attention more replaceable in some layers than others? Eg see the ideas in sandwich transformers: https://arxiv.org/pdf/1911.03864.pdf.

__UPDATE AFTER RESPONSE__

Hello authors, thanks for your response. After reviewing the updates, I'm still unconvinced. I will raise my score to a 4, but won't be recommending acceptance. I'll provide some suggestions below, but first I didn't note any strengths in my original review which was not right! So I'll start with a list of strengths.

Strengths:
- This direction of trying to understand how much value dot product self-attention adds is very interesting. Synthesizing the attention matrix, rather than computing pairwise dependencies is a cool idea.
- The experiments are on a range of tasks including machine translation, language modeling, GLUE/SuperGLUE and more.
- The performance of the random synthesizer is quite surprising, the fact that it doesn't depend on input tokens but can still achieve non-trivial performance is intriguing.

Suggestions to improve:
- I still think the paper could do a better job of reporting a more complete set of experiments/comparisons. Comparing against the variants of synthetic attention is interesting but not enough given that there are quite a few papers that investigate similar ideas. Dynamic convolutions -- Wu et al. report a range of experiments on machine translation, language modeling etc. Why not compare to them on these tasks as well? Comparing only to self-attention just isn't enough since **synthetic attention is not the first attempt to replace it**.
- It seems a bit strange that dynamic convolutions are competitive with self-attention in the original paper, but results on GLUE are so much worse. It might be worth verifying on the sequence generation tasks that results are as expected. For GLUE, Linformer has results in the original paper, why not also compare against it here?
- The paper needs some revision to clarify the motivations -- it starts out by talking about how self-attention may not be necessary, but in some of the results synthetic attention has to be combined with dot product self-attention to achieve reasonable performance. **On GLUE, looks like the deterioration from using synthetic attention alone is as large as 10 points on average.** The fact that it improves performance to use self-attention and add some parameters strategically can still be interesting I guess, but the original motivation of the paper starts to fade.
- Small note: Everywhere, that the baseline is "Transformer" that's a self-attention-only variant (V), so maybe the notation/tables could clarify that point.

---

> ### Author Response · Authors · 2020-11-21
> **Response to review**
>
> Thanks for the feedback and comments.
>
> Regarding synthetic attention, the T5 base about 229M params, while the Synthesizer (+V) variants are about 243 params. We also run experiments with a T5 model of equivalent parameters (243) and compare it (denoted as T5 base+). Results are updated in Table 5. Learned attention weights can be found in the supplementary material.
>
> Regarding dynamic convs, your main complaint here was about the T5 pretraining task MLM perplexity “not meaning much”. To this end, we fine-tuned dynamic convs on Glue/SuperGLUE and have updated results in Table 5. This is fair side-by-side apples-to-apples comparisons of Dynamic Convs and Synthesizers on 16 NLP/NLU tasks. Our results show that dynamic convs do not outperform even random synthesizers. It is good to note that dynamic convs also perform poorly because they do not have the cross attention inductive bias in the encoder - a similar fair situation with random and dense synthesizers.
>
> Regarding the fixed random Synthesizer, we have reported results in Table 2 (labelled as Fix). We agree that this could have been more obvious/clearer and have revised the paper accordingly. We explicitly explained this in a newly added “notation of variants” section.
>
> Regarding convergence, we have updated the paper (supplementary material) with convergence curves on SuperGLUE mixture. Overall, all models have pretty similar convergence to vanilla Transformer models (also observed on other tasks). Thanks for the suggestion.

---

### Official Review · AnonReviewer3 · 2020-10-28
**Review summary**

**Rating:** 5
**Confidence:** 4

**Review:**

The authors propose a new Transformer variant that removes the pair-wise dot-product attention but use pointwise non-linearity instead. Here are my comments.

Regarding Equation 2:

1. In Eqn(2), if W2 is a d \times l matrix, how can F(X) be an l \times l matrix? Should there be a transformation? There are two terms b in Eqn 2. Are they the same, and what is the shape of b?

2. It will be great if you can also present the dot-product attention equation together with your equation 2&3. That would be more helpful to the readers.

Regarding Equation 4:

1. The matrix R is randomly initialized. Is it fixed, or is optimized during training? If it is randomly initialized and fixed, Is the performance sensitive to the random seed? If it is randomly initialized and further optimized, could you show some learned patterns of this matrix R?

2. If R is independent of tokens, what is the difference between R and positional encoding (in particular, the relative positional encoding?)

3. Token-token attention can handle cases that exceed length l. For your random attention R, how do you handle length that more than l?


Regarding Experimental results:

1. The authors propose a variant called the random synthesizer. However, from the experimental results, we can see that the random synthesizer is worse than the baselines (significantly worse on GLUE benchmarks). I am not sure of the reason behind presenting this variant of networks. There is no gain but only a drop for this choice.

2. I couldn't find a place that defines what the model R+V is and what is D+V.

3. In GLUE, the WSC dataset is usually difficult. Could you explain why your model has a significant drop compared to the baseline model?

4. In both GLUE and superGLUE data, small-task performance (COLA, RTE..) is quite unstable. For large tasks, such as MNLI and QNLI, the performance improvement is not much but just comparative.


Overall comments:

Dot-product attention is one of the key components in the Transformer model, but the necessity of the dot-product attention is not very clear. The authors make a step to understand the attention mechanism. However, from both theoretical and empirical results, I cannot see any strong motivations behind the modification or any empirical benefits. It seems to me that the new model cannot be a good replacement for the original model. But I am open to further discussions.

---

> ### Author Response · Authors · 2020-11-21
> **Response to Review**
>
> Thanks for the review and insightful comments/feedback.
>
> Regarding Eqn 2, this is applied position-wise to $X_i$ and the output of $F_(X_i)$ is $\ell$. Since this is applied position-wise to all elements in the sequence, the output is a matrix of $\ell \times \ell$. Regarding b, we have fixed them (they are two different bias terms corresponds to the the two linear layers).
>
> Regarding the dot product attention equation, we have added the original DP attention equation after Eqn 3 in the revised version.
>
> Regarding the Random matrix, we tried both fixed and trained random. The fixed random results are presented in Table 2 which are on WMT EnDe, WMT EnFr and LM1B (denoted as “Fix”). The trained variation is represented as “R”. We have shown some random patterns in the supplementary material.
>
> Regarding the length beyond L, we discussed this in “On Parameters Depending on Sequence Length” of the paper (Section 3, Page 5). To handle beyond L, one can adopt R that is repeatedly tiled in blocks or we could also potentially extrapolate beyond L by simply expanding R by upsampling. The simplest method is to initialize and pretrain with a high upper bound (for safety) of L and then only use <L (or whatever the task requires) in downstream applications.
>
> **Regarding experiment results:**
>
> Random synthesizers alone do decently on all other tasks (MT, LM etc) except GLUE/SuperGLUE. The reason for this is because GLUE has many tasks that require cross-attention where the cross attention happens in the self-attention. This is a fundamental issue with cross attention and not really self-attention. Similarly, if we train a RNN-based model or CNN based model on GLUE, we will not be able to emulate cross attention. In short, the key take-away message here is that if the task is not a “text matching” task, then Random Synthesizers can do at times as well as vanilla Transformers .
>
> Dynamic Convolutions (Wu et al.) are also one example of an architecture that claims to be able to replace self-attention. We just added finetuning results of Dynamic Conv on SuperGLUE and GLUE to the revised paper. The same effect is observed - they perform poorly because of not being able to utilize cross-attention. In the same setting, it is worth noting that our Random Synthesizers can outperform Dynamic Conv (by a decent amount). [Updated Table 5 in paper]
>
> Do we claim that folks should start replacing their Transformers with Synthesizers? No. The point of Random Synthesizers is just to show that they are competitive, and this might lead to interesting advances of the Transformer based research in general. It is also worth knowing when and why dot product attention is useful (and conversely, when they are not).
>
> R and D represent Random and Dense variations respectively. V refers to a mixture model of Synthesizers + Vanilla DP attention. The R+V and D+V variations are mainly to show that synthetic attention improves Transformers - the secondary message of our paper. Thanks for the suggestion, we will make this clearer in the revised version. We have added a section of Notation of Variants in the experiment section to make this clearer.
>
> About results on Glue/SuperGLUE, all models were trained on all 16 tasks at the sametime following the T5 setup. This is perhaps how the model makes trade-offs during the training. We believe that the overall SuperGlue and Glue score is a little more indicative of the overall strength of the architecture.
>
> **Regarding overall comments:**
>
> There are mainly two key messages of this paper. The first one is that random/dense without DP is good enough for a good number of tasks (e.g., MT). Without dot product attention, Transformers actually hold up pretty well. However, while this is worth saying, this is not saying that we should just take away DP attention.
>
> The second message is that Synthetic Attention (Dense or Random) improves the Transformer for all the tasks we tried. Our experiments are pretty extensive and conducted on many datasets. The parameter costs added are pretty negligible (229->243 for T5 base) and result in a good increase on all the datasets we’ve tried.
>
> Thanks for the review!

---

### Official Review · AnonReviewer4 · 2020-10-29

**Rating:** 7
**Confidence:** 4

**Review:**

=================================

Summary

This paper challenges the common belief that self-attention with dot product is necessary to train good NLP models. Several variants of the Synthesizer model is proposed. The effectiveness of Synthesizer is surprisingly good, although not beating the dot-product attention. The authors further showed that mixing synthesizer and dot-product attention sometimes achieve better results. The idea is validated on Translation, NLU, Summarization, Dialogue, and Language Modeling.

=================================

Review

I enjoyed reading this paper and I imagine it would benefit many researchers in this community. I can't find any reason to reject this paper. However, it does not propose a new model that completely beats the transformer so I wouldn't give a higher score.

Pros
-	The idea of Synthesizer is novel and could inspire the community to rethink self-attention transformers.
-	Experiment are thorough and cover a wide range of different tasks.
-	Paper is well-written.

Cons
-	This model cannot fully replace dot-product attention, although it’s not a big problem in my opinion.

=================================

Other Questions / Suggestions

- Why not finetune from the MLM Synthesizer for GLUE? Does Synthesizer also benefit from MLM pretraining?
- What D/R/V stand for is not clearly stated in the paper.

---

> ### Author Response · Authors · 2020-11-21
> **Response to review**
>
> Thanks for the kind comments and great review! We are excited that you enjoyed our paper.
>
> Regarding questions:
>
> On the GLUE/SuperGLUE finetuning experiments, we finetuned Synthesizer and T5 (and now the new Dynamic Convolutions and T5 base+) models that have been trained using the MLM T5-pretraining objective.
>
> D/R/V stands for Dense, Random and Vanilla. We apologize for not making this clearer in the paper. We have added a section “Notation of Variants” in the revised version of the paper.
>
> Finally, the Mixture Synthesizers (R+V or D+V) variants often outperform Transformers. We show that Synthetic Attention + Transformers almost always leads to an improvement.
>
> Thanks for taking the time to review our paper and also the positive comments! We are glad that you enjoyed your paper.

---

### Author Response · Authors · 2020-11-22
**Summary of Paper Updates**

Dear Reviewers and Area Chairs,

Thanks for all the constructive feedback and spending the time to review our paper.

Please find the list of updates to our paper below:

1) We added fine-tuning results on GLUE/SuperGLUE for Dynamic Convs (Wu et al.) and show that Random Synthesizers outperform Dynamic Convs in both pretraining perplexity and finetuning (this negates a primary concern from AnonReviewer1)
2) We added a Transformer baseline (denoted T5 (base+)) which has been scaled up to match the Synthesizer (R+V) model to show that our model is **not** outperforming because of extra parameters. This is updated in Table 5/6. Again, this negates another primary concern from AnonReviewer1).
3) We added convergence curves on SuperGLUE to the supplementary material.
4) We made notations of Eqn2 and added the standard DP attention formula as requested by AnonReviewer3.
5) We added a section that describes all the different variants and what each abbreviation means. This is requested by multiple reviewers that will help with interpreting the experiment results.

Thanks.

---

### Decision · Program_Chairs · 2021-01-07
**Final Decision**

**Decision:**

Reject

**Comment:**

The paper seeks to answer the question on the necessity of the self-attention matrix in Transformers and whether it is possible to synthesize it by alternate means other than pairwise attention.
The reviewers appreciated the main general idea and the wide range of experiments conducted.
However, there are some concerns on clarity and evidence supporting main claims. While authors tried to address certain concerns through revision and response, the results suggests that self-attention matrix is still needed for strong performance and cannot be fully replaced by synthesizers. While authors also acknowledge this in discussions, If we do not consider the combined models (R+V or D+V), the empirical results do not look very convincing on the competitiveness of Synthesizers. They are only competitive on MT/Dialogue while Failing quite considerably on GLUE and Summarization. Overall, I felt very positive about the direction the paper pursues, but the empirical results doesn't seem to fully support the claims.   Quoting some points from reviewer discussions:

> `Comment: Moving towards an analytic framing would necessitate having the bare minimum set of experiments/comparisons before running additional analyses, but the bare minimum is still needed for this paper.

> Comment: I think the paper needs a round of revision and experiments need additional, carefully chosen baselines to adequately present synthetic attention in the context of existing solutions.

> Comment: this is not a reason that some random explored idea should be viewed as a great contribution, given that there are already several theory-grounded papers appeared in ICML (linear-attention..), NIPS (Linformer, follow-up work from linear-attention..), and ICLR (Random-feature attention, Performer..) this year. Compared to those theoretically well-motivated attention modification papers, this work is not that solid.